# Airway environment drives the selection of quorum sensing mutants and promote *Staphylococcus aureus* chronic lifestyle

Xiongqi Ding [1], Catherine Robbe-Masselot[2], Xiali Fu[1], Renaud Léonard[2], Benjamin Marsac[2], Charlene J. G. Dauriat[3], Agathe Lepissier[1], Héloïse Rytter[1], Elodie Ramond[4], Marion Dupuis[1], Daniel Euphrasie[1], Iharilalao Dubail[1], Cécile Schimmich[5], Xiaoquan Qin[6], Jessica Parraga[7], Maria Leite-de-Moraes[1], Agnes Ferroni[7], Benoit Chassaing[3], Isabelle Sermet-Gaudelus[1], Alain Charbit [1], Mathieu Coureuil [1] ✉ & Anne Jamet [1,7] ✉

*Staphylococcus aureus* is a predominant cause of chronic lung infections. While the airway environment is rich in highly sialylated mucins, the interaction of *S. aureus* with sialic acid is poorly characterized. Using *S. aureus* USA300 as well as clinical isolates, we demonstrate that quorum-sensing dysfunction, a hallmark of *S. aureus* adaptation, correlates with a greater ability to consume free sialic acid, providing a growth advantage in an air-liquid interface model and in vivo. Furthermore, RNA-seq experiment reveals that free sialic acid triggers transcriptional reprogramming promoting *S. aureus* chronic lifestyle. To support the clinical relevance of our results, we show the co-occurrence of *S. aureus*, sialidase-producing microbiota and free sialic acid in the airway of patients with cystic fibrosis. Our findings suggest a dual role for sialic acid in *S. aureus* airway infection, triggering virulence reprogramming and driving *S. aureus* adaptive strategies through the selection of quorum-sensing dysfunctional strains.

*Staphylococcus aureus* is a common opportunistic pathogen infecting the lungs of patients with chronic lung diseases such as cystic fibrosis (CF)[1–4]. CF is the most common life-limiting autosomal recessive disorder in the Caucasian population[5]. Respiratory epithelial cells of CF patients produce thick and sticky mucus as a consequence of defective transmembrane conductance regulator *CFTR* gene[6], which impedes bacterial mucociliary clearance[7]. CF patients are chronically infected by lung-adapted *S. aureus* clones[8–10]. *S. aureus* adaptive mechanisms include the emergence of small colony variants (SCVs)[11–13], mucoid phenotypes[14], and deregulation of toxins[10,15]. Agr (accessory gene

regulatory) system is a quorum-sensing system involved in both virulence- and metabolism-associated gene regulation[16]. The role of the Agr system in the interplay between *S. aureus* and host-derived metabolites has received less attention[16] compared to the regulation of virulence factors[17,18]. By investigating the genetic mutations acquired during the course of chronic lung infection across multiple patients with CF, we observed a convergent evolution with genetic alterations at the *agr* locus leading to quorum-sensing system dysfunction[10]. Indeed, Agr dysfunction, which is associated with increased biofilm formation, is a common adaptive strategy identified in chronic

[1]Université Paris Cité, INSERM UMR-S1151, CNRS UMR-S8253, Institut Necker Enfants Malades, F75015 Paris, France. [2]Université Lille, CNRS, UMR 8576 - UGSF - Unité de Glycobiologie Structurale et Fonctionnelle, F-59000 Lille, France. [3]INSERM U1016, CNRS UMR8104, Université Paris Cité, Team «Mucosal Microbiota in Chronic Inflammatory Diseases», F75014 Paris, France. [4]Genoscope, UMR8030, Laboratory of Systems & Synthetic Biology (LISSB), Xenome team, F91057 Evry, France. [5]Anses, Laboratory of Animal Health in Normandy, Physiopathology and epidemiology of equine diseases (PhEED), RD 675, F14430 Goustranville, France. [6]Université Paris Cité, Institut de physique du globe de Paris, CNRS, F75005 Paris, France. [7]Department of Clinical Microbiology, Necker-Enfants Malades Hospital, AP-HP Centre Université de Paris Cité, F75015 Paris, France. ✉e-mail: mathieu.coureuil@inserm.fr; anne.jamet@inserm.fr

infections of the lung, bone, and skin[19-22]. The ability of Agr variants to promote a proinflammatory response in respiratory epithelial cells[10] while also exhibiting improved stress resistance[23] could contribute to their emergence in vivo.

In the context of chronic lung infection, *S. aureus* is surrounded by airway mucus and microbiota. Mucins are heavily glycosylated proteins responsible for mucus viscoelastic properties. Approximately 80% of the molecular weight of mucins are glycans, primarily composed of N-acetylgalactosamine, N-acetylglucosamine, fucose, galactose, and sialic acid[24,25]. The action of sialidases secreted by some bacterial species allows the release of sialic acid from mucins[26]. Although the genomes of *S. aureus* do not code sialidases, it can be assumed that *S. aureus* may rely on sialidases released by other bacterial species present in the lungs, such as streptococci or anaerobes[27], which could make sialic acid bioavailable. Indeed, the presence of genes encoding exo-alpha-sialidase activity within the lung microbiota has been shown by shotgun metagenomic sequencing[28,29]. Furthermore, an increased sialylation of the airway mucins in the context of CF has been described[30,31], which led us to hypothesize that sialic acid could select for adapted variants with an increased ability to use this energy source in the airways.

The purpose of this study was to investigate the impact of sialic acid, which is highly abundant in the lung environment, on *S. aureus* and, in turn, how bacterial adaptation to the lung impacts *S. aureus* and its interactions with sialic acid (Fig. 1A). We demonstrated that lung-adapted variants with an Agr dysfunction have an increased ability to use sialic acid resulting in a growth advantage both in vitro and in vivo. We further showed that sialic acid in the *S. aureus* airway environment triggers a chronic virulence program. Our results also suggest that sialic acid upregulates the iron acquisition-associated *sbn* locus, through NanR and the small RNA *isrR*. To support the clinical relevance of our findings, we showed the co-occurrence of *S. aureus*, sialidase-producing microbiota, and free sialic acid in the airway of patients with CF. Overall, our results suggest that free sialic acid, widely available from airway mucins, is potentially an important metabolic factor involved in transcriptomic reprogramming and in the selection of Agr variants, thereby promoting long-term chronic infection.

## Results

### Agr system dysfunction correlates with sialic acid hyperconsumption and is selected in an environment containing free sialic acid

*S. aureus* can uptake and metabolize sialic acid (or Neu5Ac), a 9-carbon monosaccharide thanks to the *nan* locus that encompasses 5 genes (*nanA*, *nanK* and *nanE* encoding enzymes involved in the catabolism and *nanR* and *nanT* encoding regulator and permease respectively)[32] (Fig. 1B). The *nan* locus is repressed by NanR in the absence of sialic acid, whereas in presence of sialic acid, the Neu5Ac catabolic intermediate *N*-acetylmannosamine-6-phosphate (ManNAc-6p) relieves NanR promoter binding allowing *nan* locus expression[32]. Agr system inactivation has been previously correlated with upregulation of the *nan* locus in MW2 and USA500 reference strains in undefined media[33,34]. We thus sought to determine whether the increased expression of *nan* locus could drive *S. aureus* adaptive strategy in chronic airway infection associated with Agr dysfunction. For control purposes, we first built a Δ*nanE* mutant unable to catabolize sialic acid and its corresponding complemented strain showing a direct link between sialic acid consumption and bacterial growth in a defined medium with Neu5Ac supplementation (Fig. S1A, B). In this medium, the expression of the *nan* locus was compared in a pair of reference strains (USA300-LAC and its *agrC* defective derivative strain Δ*agrC*) confirming an upregulation of *nan* gene expression in Δ*agrC* derivative compared to the parental strain (Fig. 1C). We also tested in prototypical pairs of early-late lung-adapted clinical isolates that we have previously characterized[8,10,35] in which only the latest isolates exhibit

Agr dysfunction (CF1_late and CF16_late isolates). In CF1_late isolate, a premature stop codon has been identified in *agrC* (L193Stop), and in CF16_late isolate, a premature stop codon has been identified in *agrD* (L16Stop). A lack of cytotoxicity (assessed by LDH release assay) and a decreased RNAIII expression (assessed by qRT-PCR) have been reported in late isolates of CF1 and CF16 compared to early isolates[10]. An upregulation of *nan* gene expression was recorded in the two prototypical Agr dysfunctional CF1_late and CF16_late isolates compared to cognate early isolates (Fig. 1D and Fig. S1C).

We then monitored the kinetics of consumption of sialic acids in the culture supernatant of each strain. We observed that Δ*agrC* deletion mutant and prototypical clinical Agr dysfunctional variants consumed sialic acid faster compared to their respective parental strains (Fig. 1E, F and Fig. S1D). Altogether, these data showed that Agr system dysfunction, selected during chronic airway infection, is associated with an overexpression of *nan* genes, which results in an increased ability to utilize sialic acid. Since Agr dysfunctional strains were able to consume free sialic acid more quickly than parental strains, we next postulated that they might be favored in a medium containing free sialic acid and could outcompete the Agr wild-type population. We first checked the growth of monocultures of wild-type and dysfunctional Agr strains and showed similar growth in the media regardless of the added carbon source (glucose, Neu5Ac, or glycerol) (Fig. S2A).

Then, we performed a competition assay with USA300-LAC (hemolytic and erythromycin-susceptible) and Δ*agrC* derivative mutant (non-hemolytic and erythromycin-resistant) and monitored the proportion of non-hemolytic variants (a proxy to quantify Agr dysfunctional variants) and erythromycin-resistant colonies in the bacterial population in presence or absence of sialic acid during 5 days. We observed that from the third day of the experiment, non-hemolytic variants represent the majority of the population when sialic acid is present in the environment, in contrast to their near absence in a medium containing glucose or glycerol (Fig. 1G, H and Fig. S2B, C). The results of selective plating on antibiotics showed the same ratios of erythromycin-resistant/susceptible colonies compared to non-hemolytic/hemolytic colonies and suggested that non-hemolytic colonies almost exclusively correspond to the *agrC* mutant initially inoculated (Fig. S2D). Of note, no non-hemolytic variant was detected in the 5-day control experiment, where only the USA300-LAC strain was initially inoculated in the presence of glucose or sialic acid, suggesting that sialic acid does not impose a pressure leading to the emergence of novel dysfunctional Agr variants. To check for the appearance of mutations during the course of the 5-day-competition experiment, we sequenced the genome of 10 non-hemolytic isolates and 10 hemolytic isolates obtained at day 5 in the presence of Neu5Ac taken from 2 independent experiments. We did not find any convergent mutations present only in the *agrC* strain compared to the wild-type strain that would explain the selection of *agrC* variants (Supplementary Data 1). Thus, the drop observed on day 1 of the competition assay suggests that the *agrC* mutant may adapt to the presence of Neu5AC in the medium before becoming predominant.

### Co-occurrence of free sialic acid, sialidase-producing bacteria, and *S. aureus* in CF sputa

*S. aureus* species does not possess a sialidase gene and potentially relies on sialidase-producing species of the surrounding microbiota to acquire free sialic acid through a dedicated transporter (encoded by *nanT*). Our comparative analysis of the O-glycan content of native mucins and mucins in the presence of USA300-LAC, using matrix-assisted laser desorption/ionization mass spectrometry, further demonstrated that *S. aureus* was unable to cleave sialic acids from mucins. Indeed, around 86% of O-glycans were sialylated in native mucins and the same level of sialylation (around 88%) was observed after incubation of mucins with USA300-LAC (Fig. S3). It is known that sputum proteins are oversialylated in CF compared with non-CF[36,37].

One can anticipate that the co-occurrence of sialylated mucins and sialidase-producing species is likely to result in the presence of free sialic acid in the airways. However, it has to our knowledge, not been specifically evaluated in the lungs of CF children. Thus, as proof of concept, we decided to verify the availability of free sialic acid and sialidase-producing species in the airway of a limited series of 9 CF children. We assessed the amount of free sialic acid in sputa in which routine microbiology analysis has identified *S. aureus* (5 patients) or not (4 patients). We detected free sialic acid, although at

heterogeneous concentrations (mean 0.12 mM, range 0.04–0.28 mM), in all CF patients tested (*S. aureus* positive: Fig. 2A; *S. aureus* negative: Fig. S4A). In parallel, we investigated by 16 S amplicon and shotgun metagenomic analysis, the microbiota composition of the sputa of the 5 CF children with *S. aureus* (Fig. 2B and Fig. S4B). This analysis showed the presence of species for which sialidase activity has been previously demonstrated[38–41]. Indeed, all the patients harbored at least one of the following species: *Schaalia odontolytica* (previously known as *Actinomyces odontolyticus*), *Prevotella melaninogenica*, or *Streptococcus* mitis

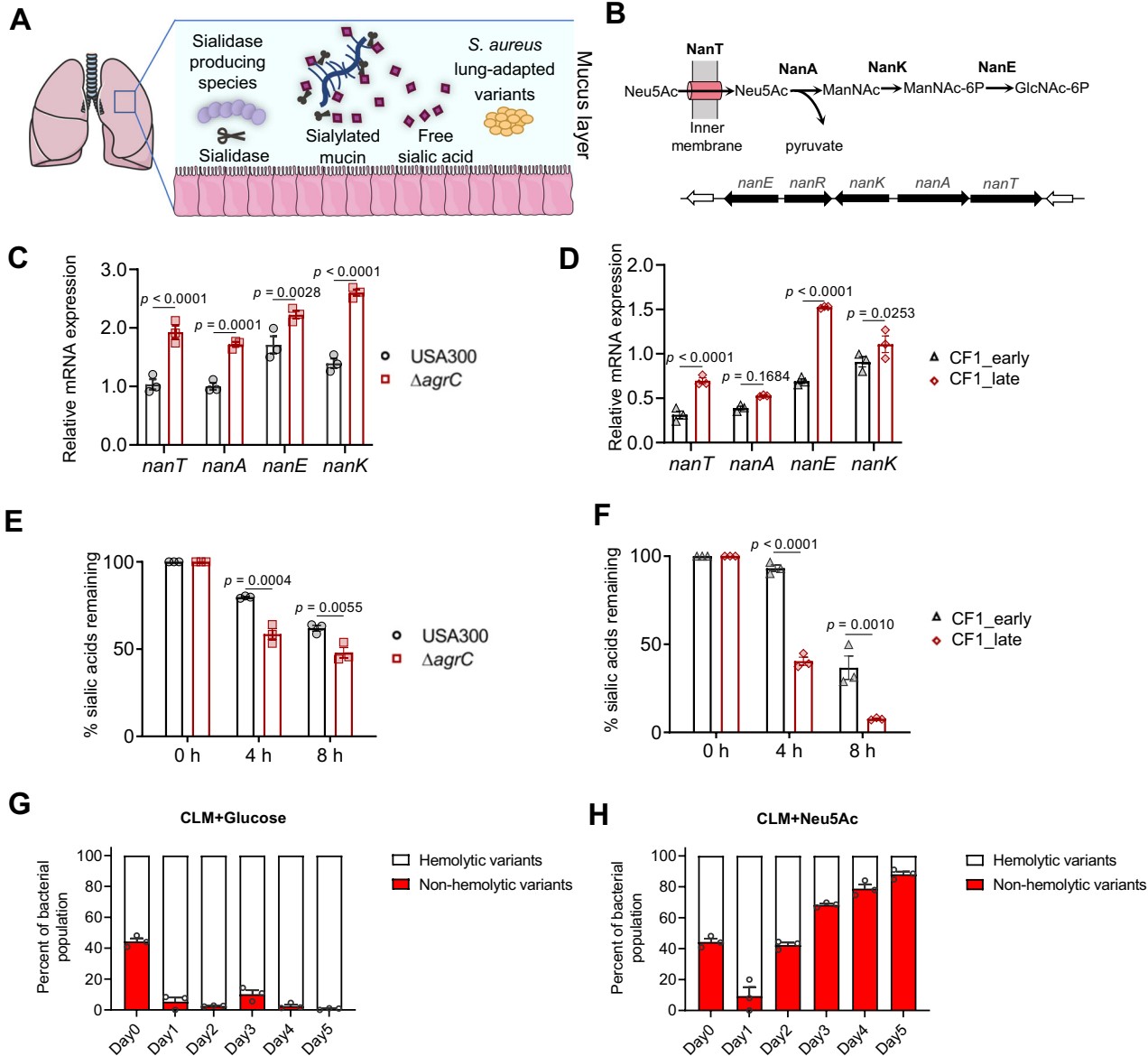

**Fig. 1 | Agr system dysfunction correlates with sialic acid hyperconsumption and is selected in an environment containing free sialic acid. A** Schematic depicting the CF lung environment encountered by *S. aureus* lung-adapted variants with co-occurrence of sialylated mucins and sialidase-producing bacterial species. **B** Schematic depiction of *nan* locus and sialic acid metabolism pathway. **C, D** *nan* locus expression from bacteria grown in CLM supplemented with 1.3 mM Neu5Ac (CLM+Neu5Ac) was quantified by qRT-PCR relative to that of housekeeping gene *gyrB* (OD600nm - 0.6) (*n* = 3 samples/group). **C** USA300-LAC (USA300) strain and Δ*agrC* derivative; **D** Early (CF1_early) and late (CF1_late exhibiting an *agrC* inactivation) isogenic isolates recovered 3 years apart from respiratory samples of cystic fibrosis patient CF1. **E, F** Kinetics of consumption of sialic acids obtained from chemical desialylation of bovine submaxillary mucins (BSM) (*n* = 3 samples/group). Bacteria were grown in CLM supplemented with 0.2% chemically desialylated BSM.

Free sialic acids were derivatized with DMB (1,2-diamino-4,5-methylene dioxybenzene) before quantification by HPLC. **E** USA300-LAC (USA300) strain and Δ*agrC* derivative; **F** Early (CF1_early) and late (CF1_late) isogenic isolates. **G, H** Agr dysfunctional variant monitoring in a competition assay. USA300-LAC (USA300) strain was completed for 5 days with Δ*agrC* derivative starting at a ratio of 1:1 in the presence of **G** 1.3 mM glucose or **H** 1.3 mM Neu5Ac. Proportions of nonhemolytic and hemolytic subpopulations were determined based on colony phenotype on sheep blood agar (*n* = 3 samples/group). The phenotype of USA300 and Δ*agrC* colonies is hemolytic and non-hemolytic, respectively. Error bars indicate mean with SEM. Statistically significant differences were calculated by one-way ANOVA with Bonferroni's multiple comparisons test and *p* value was indicated. Source data are provided as a Source Data file.

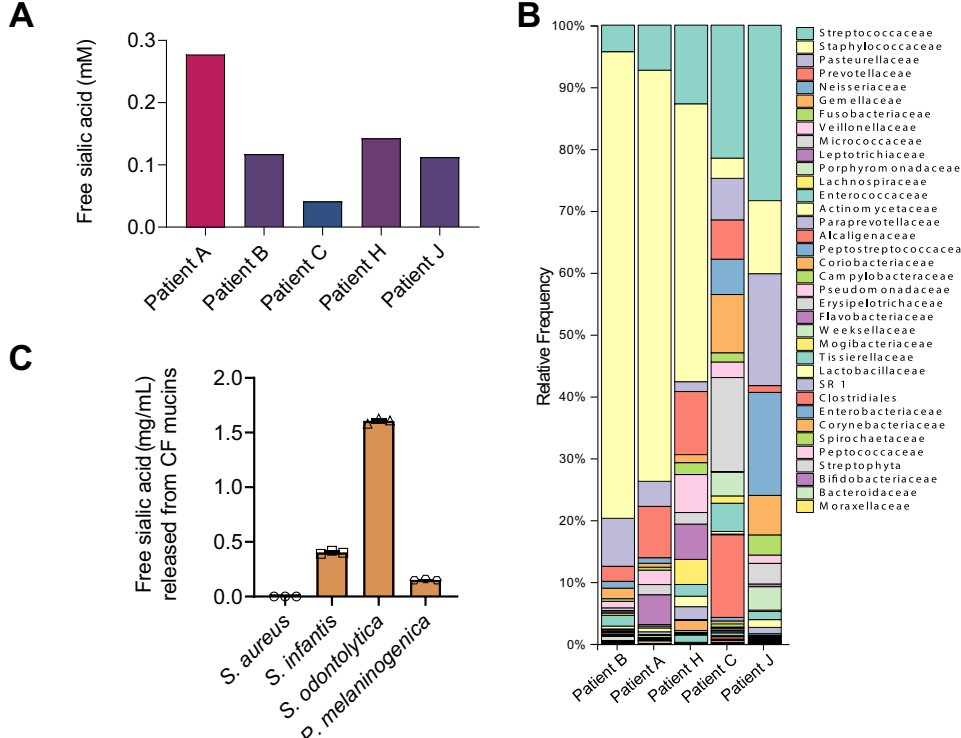

**Fig. 2 | Co-occurrence of free sialic acid, sialidase-producing bacteria, and *S. aureus* in CF sputa. A** Free sialic acid concentration has been assessed by colorimetric assay in sputa from five children with CF and positive *S. aureus* culture detection by standard microbiology testing. **B** Taxonomic characterization at the family level of microbiota in the sputa of 5 patients with CF and positive *S. aureus* culture detection showing the concomitant presence of Staphylococcaceae, Streptococcaceae, and Prevotellaceae families by 16 S sequencing. **C** The sialidase activity of three isolates obtained in culture (*Streptococcus infantis*, *Schaalia odontolytica* and *Prevotella melaninogenica* from sputa of Patient A, Patient C, and Patient H, respectively) was evaluated by quantifying the amount of salic acid released from purified CF patient mucins. The USA300-LAC strain was used as a negative control. Source data are provided as a Source Data file.

group species (*S. mitis, S. oralis*, or *S. infantis*). In addition to species identification, shotgun metagenomics allows the detection of fragments of sialidase genes in 3 sputa (Patients B, C, and H). However, due to the high number of human cells within CF sputa, shotgun metagenomics is not a sensitive technique for the identification of specific bacterial genes. Therefore, to directly prove that isolates from CF sputa displayed functional sialidase activity, we isolated 3 bacterial strains from CF sputa belonging to 3 distinct representative species (*S. infantis* from patient A, *S. odontolytica* from patient C, and *P. melaninogenica* from patient H) and we directly confirmed their ability to release free sialic acid from mucins obtained from respiratory secretion of CF patients (Fig. 2C). In contrast, the control USA300-LAC strain was not able to release free sialic acids (Fig. 2C). We also investigated patients without *S. aureus* in their sputum by 16 S amplicon metagenomic analysis (Fig. S4C) and confirmed the systematic presence of species belonging to the *Streptococcaceae* family. In the four patients without *S. aureus*, we isolated strains of *Streptococcus mitis* group species (*S. oralis*) by culture and directly evidenced their ability to release free sialic acid from CF airway mucins (Fig. S4D). Collectively, these data directly showed that sialidase-producing species were present in CF sputa regardless of the presence of *S. aureus*.

### Free sialic acid provides a growth advantage to Agr dysfunctional variants in vitro and in vivo

To simulate the lung environment, we used respiratory epithelial cells cultured at air–liquid interface (ALI model as described previously[10,42]). We did not detect free sialic acid in germ-free mucus produced by Calu-3, whereas 24 hours of mucus treatment with a bacterial sialidase, simulating microbiota-producing sialidase, led to the detection of a significant amount of free sialic acids (Fig. 3A). In addition, we

sequenced an *S. mitis* clinical isolate[42] and identified a sialidase gene in the annotated genome. We further showed the ability of the *S. mitis* strain to release free sialic acid from airway mucus in the ALI model (Fig. 3A).

USA300-LAC, Δ*agrC*, and clinical prototypical strains were tested in the Calu-3 ALI model for their ability to grow after addition of sialidase or free sialic acid (Fig. 3B). Inactivation of Agr system was associated with a greater increase in CFU numbers upon addition of sialidase (CFU fold change of 1 versus 5 for the Δ*agrC* derivative compared to parental strain and CFU fold change of 1 versus 6 for CF1_late compared to CF1_early isolate) (Fig. 3C). A similar result was observed after the direct addition of free sialic acid (CFU fold change of 3 versus 24 for the Δ*agrC* derivative compared to parental strain and CFU fold change of 8 versus 50 for CF1_late compared to CF1_early isolate) (Fig. 3C). To better mimic the CF lung environment, we also carried out the ALI experiments with a human CF bronchial epithelial cell line with the CFTR mutation (homozygous F508del CFBE41o-) and we obtained results similar to those with Calu-3 (Fig. S5). Thus, in the ALI model, free sialic acid either directly added or released from mucus by bacterial sialidase provided a significant growth advantage to strains with Agr dysfunction compared to parental strains.

We next used a murine pulmonary infection model, where mice were intranasally infected with equal doses of USA300-LAC or Δ*agrC* derivative with or without sialic acid supplementation. Twenty-four hours after infection, mice were sacrificed and bacterial CFUs from the lungs were determined by plating serial dilutions of the tissue homogenate (Fig. 3D). Similar amounts of USA300-LAC were recovered from the lungs with or without sialic acid supplementation, whereas the bacterial load of the Δ*agrC* derivative strain was significantly increased when sialic acid was administered, therefore, demonstrating that

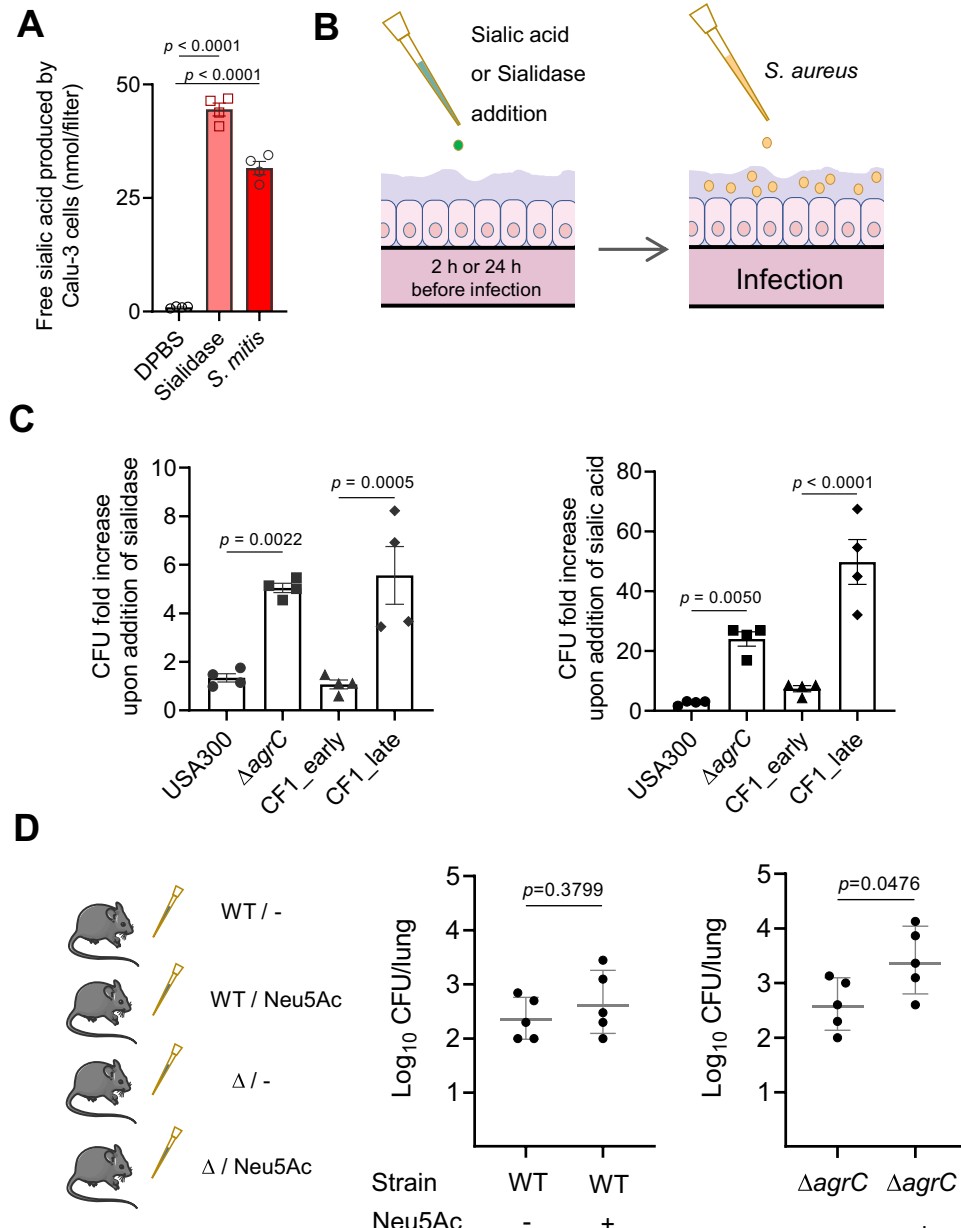

**Fig. 3 | Free sialic acid provides a growth advantage to Agr dysfunctional variants in vitro and in vivo. A** Free sialic acid in mucus produced by Calu-3 epithelial cells grown at the air-liquid interface (ALI) was measured 24 h after addition at the apical side of Dulbecco's phosphate-buffered saline (DPBS), sialidase or *S. mitis*. A statistically significant difference was calculated by two-sided unpaired *t* test and *p* value was indicated (*n* = 4 samples/group). **B** Schematic depiction of infection in the ALI culture model. **C** In Calu-3 ALI model, left panel: 5 μL sialidase (0.5 U/mL DPBS/0.1% BSA) or 5 μL DPBS/0.1% BSA was added 24 hours before infection. Right: 5 mL Neu5Ac (0.4 g/L) or 5 mL H₂O was added 2 h before infection with *S. aureus*. ALI mucus of wells was apically infected with USA300, Δ*agrC*, or early and late isolates from patient CF1. Mucus was collected 16 h post infection and plated for CFU counting. Data are represented as fold increases over wells infected with the same strains without sialidase/sialic acid addition (*n* = 2 wells/groups). Each condition was performed with two biological replicates and two technical replicates. Statistically significant differences were calculated by one-way ANOVA with Bonferroni's multiple comparisons test and *p* value was indicated. **D** Mice were inoculated intranasally with bacterial suspension (2 × 10⁸ CFUs) of *S. aureus* USA300 or Δ*agrC*, in PBS (−), or free sialic acid (+), as indicated on the bottom of each panel (*n* = 5 mice/group). Twenty-four hours post infection, the bacterial CFUs in the lungs were counted. Statistically significant differences were calculated by a two-sided unpaired *t* test and *p* value was indicated. Error bars indicate mean with SEM. Source data are provided as a Source Data file.

dysfunction of the Agr system resulted in a growth advantage in the lung.

## Sialic acid triggers a transcriptomic reprogramming favoring chronicity and iron acquisition in *S. aureus*

To gain a broader understanding of the impact of free Neu5Ac on *S. aureus*, we next performed transcriptomic profiling of USA300-LAC grown either in the presence of glucose or Neu5Ac using RNA-seq.

Importantly, pH was adjusted to ensure that pH variations did not explain the differential gene expression profile. Eighty-five genes encoding proteins (56 upregulated and 29 downregulated) were differentially expressed (equal or greater than 2 log2-transformed fold change; *p* < 0.05) between both conditions (Fig. 4A and Supplementary Data 2). Based on their function and known regulation, these genes were clustered in 6 groups linked to energy metabolism, translation, and iron uptake (Fig. 4B). As expected, we observed a drastic

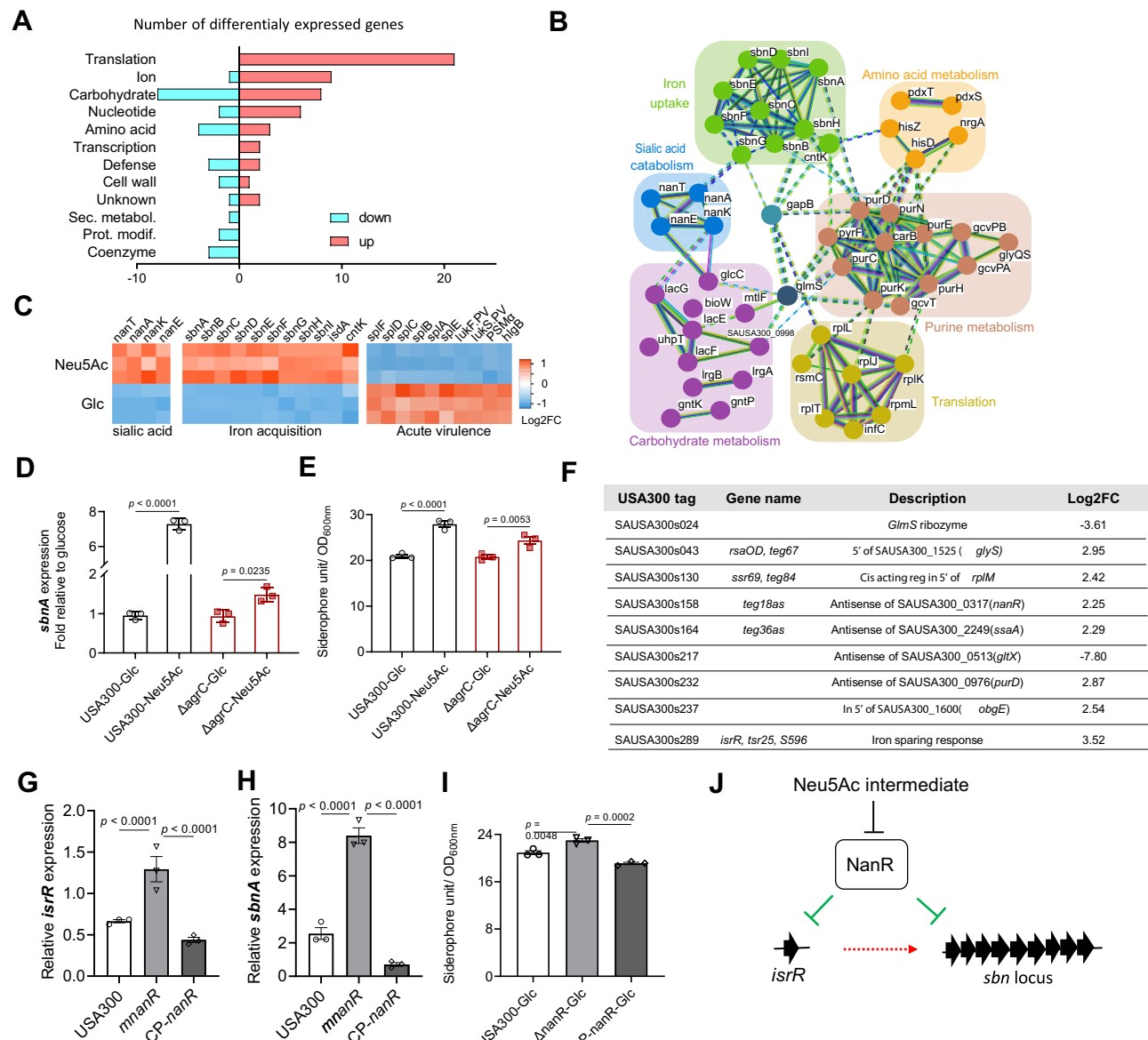

**Fig. 4 | Sialic acid triggers a transcriptomic reprogramming favoring chronicity and iron acquisition in *S. aureus*. A** RNAseq was performed to compare gene expression of the USA300-LAC strain grown in CLM with supplementation with glucose or Neu5Ac (1.3 mM). Numbers of genes with increased (red) or decreased (blue) expression levels between glucose and Neu5Ac growth conditions considering statistically significant genes with a Log2(FoldChange) ≥ |2| and FDR-adjusted *p*-value ≤ 0.01. Categorization is based on the Clusters of Orthologous Groups (COG) functional classification. Ion, inorganic ion transport and metabolism; Sec. metabol., secondary metabolites biosynthesis, transport and catabolism; Prot. modif., posttranslational modification, protein turnover, chaperones. **B** Differentially expressed genes were analyzed via the Search Tool for the Retrieval of Interacting Genes/Proteins (STRING) to identify protein–protein interaction networks. **C** Heatmap showing three biological replicates for selected genes with significant differences between glucose and Neu5Ac growth conditions. **D** USA300 and Δ*agrC* were cultivated in CLM supplemented with glucose or Neu5Ac. The *sbnA* expression (OD600 nm - 0.6) were quantified by qRT-PCR with reference to *gyrB* (*n* = 3 samples/group). Fold change was calculated relative to the glucose condition. (E) The iron acquisition ability was measured in the presence of glucose or Neu5Ac. Strains were cultured in CLM supplemented with glucose or Neu5Ac, culture supernatants were collected at post-exponential phase and the siderophore units were measured by chrome azurol S (CAS) assay (*n* = 3 samples/group). **F** List of differentially expressed small RNAs between glucose and Neu5Ac growth conditions considering statistically significant genes with a Log2(FoldChange) ≥ |2| and FDR-adjusted *p*-value ≤ 0.01. **G, H** USA300, Δ*nanR* deletion mutant, and complemented *nanR* strain (CP-*nanR*) were cultivated in CLM supplemented with glucose. *isrR* (**G**) and *sbnA* (**H**) expression (OD600nm - 0.6) were quantified by qRT-PCR with reference to *gyrB* (n = 3 samples/group). **I** The iron acquisition ability of USA300, Δ*nanR* deletion mutant, and complemented *nanR* strain (CP-*nanR*) was measured in the presence of glucose by CAS assay (*n* = 3 samples/group). **J** Schematic depicting putative pathways for NanR-dependent regulation of iron acquisition-related genes. Error bars indicate mean with SEM. Statistically significant differences were calculated by one-way ANOVA with Bonferroni's multiple comparisons test and *p* value was indicated. Source data are provided as a Source Data file.

increased expression of *nan* locus genes involved in the transport and catabolism of sialic acid (~8- to 30-fold) (Fig. 4C). In addition, the entire *sbn* locus (*sbnABCDEFGHI*), which encodes the transport, export and biosynthesis pathway of staphyloferrin B, was also highly upregulated (~6- to 30-fold) (Fig. 4B and 4C). Of interest, Neu5Ac

downregulated the expression of important virulence genes encoding Spl proteases (~4-fold)[43], Panton-Valentine leukocidin components (~3- to 4-fold)[44], phenol-soluble modulins PSMs (~4- to 5-fold)[45], and gamma-hemolysin (HlgCB) (~2.8-fold)[46] (Fig. 4C). In contrast, *agr* genes were not significantly deregulated (Supplementary Data 2).

To rule out the possibility that the lack of glucose rather than the presence of Neu5Ac could be the trigger of observed deregulations, we checked the transcript levels of *psmα* and *sbnA* genes by qRT-PCR in CLM supplemented with either glycerol, glucose, or Neu5Ac. We found that the *psmα* transcript was only downregulated in the presence of Neu5Ac, and the *sbnA* transcript was only upregulated in the presence of Neu5Ac (Fig. S6).

The induction of the expression of the whole *sbn* locus in wild-type USA300-LAC in the presence of sialic acid is one of the main findings of the RNAseq experiment. Therefore, we investigated if sialic acid also impacts *sbn* expression in the *agr* mutant background and showed that sialic acid also induced an increase in *sbn* expression (Fig. 4D). By using a phenotypic assay, we confirmed that changes in *sbn* expression in WT and *agr* mutants correlated with increased siderophore activity in the presence of sialic acid compared to glucose (Fig. 4E).

### Sialic acid reprogramming of small RNAs suggests a link between NanR repressor and iron acquisition

We also analyzed the impact of Neu5Ac on small RNAs that have been annotated in USA300 genome[47], and identified 9 sRNAs (7 upregulated and 2 downregulated) that were differentially expressed (Fig. 4F). Six sRNAs were predicted to regulate genes encoding proteins also identified in the RNA-seq (*nanR, ssaA, glyS, rplM, purD,* and *glmS*). Most interestingly, we identified the sRNA *isrR*, a major contributor to the iron-sparing response[48], also known as *S596*[49], *tsr25*[47], and *srn_2975*[50]. We excluded the possibility of iron contamination introduced by the addition of different carbon sources, showing that the concentration of iron in CLM supplemented with either glucose or Neu5Ac was comparable (493 µg/L and 555 µg/L, respectively). We also confirmed by qRT-PCR that the upregulation of *isrR* is indeed due to Neu5Ac (Fig. S6C).

A binding motif of the NanR sialic acid metabolic pathway repressor was predicted upstream of the sRNA *isrR*[50]. Furthermore, *sbn* genes were among the most positively co-expressed genes with *isrR*[50], leading us to hypothesize that NanR could regulate the expression of *sbn* locus through the regulation of *isrR*. We generated an isogenic Δ*nanR* deletion mutant in USA300-LAC background and its complemented strain (designated CP-*nanR*). Both strains displayed a growth similar to parental strain when grown in CLM with or without Neu5Ac over a 24-h period (Fig. S7A, B). We then assessed the expression of *isrR* in USA300-LAC, Δ*nanR*, and CP-*nanR* strains in CLM containing glucose or Neu5Ac (Fig. 4G and S7C). In both media, *isrR* was strongly upregulated in the *nanR* deletion mutant compared to USA300-LAC and complemented strain. The expression of *sbnA* had the same pattern (Fig. 4H and S7D). In line with *sbn* locus over-expression, the siderophore activity of *nanR* mutant compared to the parental strain was increased (Fig. 4I).

Altogether, these findings suggest that NanR could repress the sRNA *isrR*, and that the abrogation of NanR repression (due to the presence of Neu5Ac or in the context of *nanR* deletion) allows transcription of *isrR*, thereby promoting the expression of *sbn* locus and increasing siderophore activity.

### Discussion

We demonstrated for the first time, both in vitro and in vivo, that free sialic acid provides a growth advantage to Agr dysfunctional variants. Furthermore, we showed that the wild-type population is outcompeted by Agr dysfunctional variants in the presence of sialic acid. One possible explanation is that sialic acid in the airway environment could favor the accumulation of Agr dysfunctional variants through their improved capacity to use sialic acid as a carbon, nitrogen and energy source. The high frequency of Agr dysfunctional variants in CF chronic lung infection has been well described and can be considered a hallmark of *S. aureus* adaptation. The inactivation of the Agr system is

directly associated with increased adhesin expression and biofilm formation[51], which contribute to explaining the ability of Agr dysfunctional variants to establish chronic infections[20,52]. Our work suggests that an environment with free sialic acid could also help explain the selection of Agr dysfunctional variants in the lungs. Indeed, Agr dysfunctional variants exhibited an upregulation of *nan* locus correlated with an increased ability to consume sialic acid. Degradation of sialic acid into GlcNAc6P feeds the glycolytic cycle, hence providing nutritional advantage. However, the exact molecular mechanism explaining the upregulation of *nan* genes in Agr dysfunctional strains remains to be elucidated[33,34].

*S. aureus* is able to use free sialic acid in the airway thanks to the sialidase-producing species, which are commonly found in the CF lung microbiota. Another example of 'cross-feeding' contributing to airway disease has been described between *S. aureus* and anaerobic bacteria degrading mucins, thus providing mucin-derived metabolites, including amino acids usable as nutrient sources[27]. *Pseudomonas aeruginosa* and *S. aureus* also frequently coexist in CF lung microbiota, and their interaction ranges from competitive to cooperative during the course of the co-infection[53,54]. *P. aeruginosa* strains are able to secrete various virulence factors directly inhibiting the growth of *S. aureus* (e.g., staphylolysin and pyocyanin). However, since antagonistic behavior was not conserved in CF-adapted strains of *P. aeruginosa*[55,56], it appears that the relationship between *P. aeruginosa* and *S. aureus* could transition from competition to coexistence[57]. Indeed, *P. aeruginosa* is able to use acetoin produced by *S. aureus* as an alternative carbon source, illustrating trophic cooperation between both pathogens[57].

Mucin glycans have been shown to attenuate the virulence of the opportunistic pathogen *P. aeruginosa*[58]. Accordingly, our RNA-seq result showed that sialic acid downregulated some acute virulence genes, such as *spl, lukSF-PV, psmα,* and *hlg*[48]. The serine protease-like proteins (Spls) are secreted proteases involved in lung dissemination of *S. aureus* in a rabbit model of pneumonia[43] and in allergen-induced airway inflammation[59]. Panton-Valentine leucocidin (PVL) is a two-component (LukS-PV and LukF-PV) β-barrel pore-forming toxin targeting neutrophils and macrophages[18]. PVL is associated with pleuropneumonia in young children and is considered an independent risk factor for mortality in older patients with staphylococcal necrotizing pneumonia[60]. The amphipathic peptide family, which includes PSMs, plays a number of essential functions in the pathogenesis of *S. aureus*, such as cell lysis, biofilm development, and immunological regulation[45,61]. The γ-hemolysin HlgAB is able to lyse human erythrocytes, neutrophils, and macrophages[61]. Of note, *spl, lukSF-PV, psmα,* and *hlg* acute virulence genes are under the positive regulation of the Agr system[62]. However, the *agr* locus is not deregulated in the presence of free sialic acid, and the molecular mechanism explaining the deregulation of virulence genes in the presence of sialic acid requires further investigation.

In contrast, the presence of sialic acid highly upregulated expression of the entire *sbn* locus, which encodes the transport, export, and biosynthesis pathway of staphyloferrin B siderophore. The battle for iron acquisition is mediated through the production of bacterial siderophores[63] to combat the nutritional restriction imposed by host iron sequestration[64]. Remarkably, environments such as mucosal surfaces are both rich in free sialic acids and poor in free iron[65,66]. Generally, low iron levels in the microenvironment correlate with up-regulation of iron-acquisition gene expression, which are otherwise repressed by the iron-dependent repressor Fur (ferric-uptake regulator)[67–69]. We demonstrated the ability of sialic acid to upregulate staphyloferrin B (*sbn* locus) and sRNA *isrR* expression independently of iron concentration. Upregulation of both loci was also observed in *nanR* deletion mutant, which suggests a role of NanR in coordinating sialic acid and iron metabolism pathways. A putative NanR binding site has been identified upstream of the small RNA *isrR* but not upstream of the *sbn* locus favoring the hypothesis of an

indirect regulation of *sbn* via *isrR*. The regulatory network involved in the co-expression of *sbn* and *isrR* loci and the role of NanR warrant further investigations (Fig. 4J).

In this study, we used a wild-type murine model of lung infection, which imperfectly mimics human lung[70]. Indeed, the mucus in a healthy mouse lung is thin and organized as discontinuous clouds[71]. Therefore, a direct addition of sialic acid in murine airways was required as a proof of concept that sialic acid is sufficient to promote an increase in the bacterial burden of Agr variants.

Sialic acid in the airway environment could promote a chronic lifestyle and hence contribute to the selection of quorum-sensing dysfunctional variants. The pleiotropic effect of Agr dysfunction suggests that other metabolites in the airway environment may also contribute to the success of dysfunctional quorum-sensing variants. Elucidation of the full range of selection factors for quorum-sensing dysfunction during chronic infections remains a challenge for future studies.

## Methods

### Study approval
All experiments were performed in accordance with the guidelines and regulations described by the Declaration of Helsinki and the law Huriet-Serusclat on human research ethics, and informed consent was obtained from all participating subjects. The study involving human research participants was approved by Assistance Publique–Hôpitaux de Paris (AP-HP) and validated by the organization Ile-de-France 2 IRB (ID-RCB/Eudract: 2016 A00309-42). Clinical isolates of *S. aureus* were obtained from airway secretions from patients with cystic fibrosis at the Necker-Enfants Malades University Hospital, Paris, France. Sputum sampling is part of routine standard care. The animal experiments were approved by the local ethical comity of Institut Necker-Enfants Malades and the procedure was approved (APAFIS #35546-2021111714496281).

### Laboratory strains, clinical isolates, cell lines, and culture conditions
Strains are listed in Table S1. The epidemic clone *S. aureus* USA300-LAC (designated wild type [WT]) and the *S. aureus* USA300-LAC Δ*agrC* (SAUSA300_1991, clone NE873) were provided by the Biodefense and Emerging Infections Research Resources (BEI). *Escherichia coli* strain DH5a was used as a host strain for plasmid constructions. *S. aureus* strain RN4220 was used as the recipient (restriction-negative, modification-proficient) in the electroporation of the constructed plasmids. Clinical isolates of *S. aureus* obtained from airway secretions from patients with cystic fibrosis were previously described[10]. *Streptococcus mitis* strain B26E10 from the Necker Hospital collection has been previously described[42]. *Staphylococci* were grown in Brain Heart Infusion (BHI) Broth or Agar (1.5%) at 37 °C. *E. coli* was cultivated in Luria–Bertani (LB) broth or Agar. *S. mitis* strain was grown on chocolate agar polyvitex plates at 37 °C. A modified carbon-limiting medium (CLM) containing 0.25% casamino acids were used for assessing carbon source utilization[32]. CLM was further supplemented with 1.3 mM glucose or 1.3 mM Neu5Ac (#MA00746, Biosynth Carbosynth, Compton, UK). Kanamycin and erythromycin were used at 50 mg/mL and 25 mg/mL, respectively.

The human Calu-3 cells were provided by the American Type Culture Collection (ATCC® HTB-55™, Manassas, VA, USA) and maintained in Opti-MEM™, supplemented with 5% Fetal bovine serum (FBS), 1% HEPES and 1% Non-Essential Amino Acids (#31985047, #10270106, #15630056 and #11140050, Thermo Fisher Scientific, Waltham, MA, USA).

F508del Cystic Fibrosis Bronchial Epithelial immortalized cell lines (F508del homozygous CFBE41o-) were generously provided by Dieter Gruenert (University of California, San Francisco). Cells were cultured in Minimum Essential Medium with glutamine (MEM, #31095029, Thermo Fisher Scientific, Waltham, MA, USA) supplemented with 5% FBS.

Both cell lines were grown in a humidified 5% $CO_2$ incubator at 37 °C.

### Competition experiments and sequencing of isolates on day 5
At Day 0, *S. aureus* USA300-LAC and Δ*agrC* were mixed in a 1:1 ratio with an initial $OD_{600nm}$ of 0.05 in CLM supplemented with 1.3 mM glucose, Neu5Ac or glycerol. Every 12 h during 5 days, the culture was transferred (1% v/v) to fresh media. The cultures were serially diluted and plated daily on Columbia agar supplemented with 5% sheep blood (bioMérieux) and BHI agar containing erythromycin to determine the proportion of nonhemolytic/ hemolytic variants and erythromycin-resistant/susceptible colonies within the population. On day 5, we randomly selected 5 hemolytic and 5 non-hemolytic isolates for whole-genome sequencing in 2 independent experiments. Libraries were prepared using NexteraXT Illumina kit and sequenced on a MiniSeq Illumina instrument (2 × 150 bp). USA300-LAC and Δ*agrC* parental strain genomes were assembled with unicycler[72] and annotated with prokka v1.11[73] to be used as references by snippy v3.1 (https://github.com/tseemann/snippy) to identify nucleotide polymorphisms.

### RNA extraction and qRT-PCR
Bacteria were collected at post-exponential growth phase. Pellets were resuspended in 50 μL of TE buffer containing 10 μg of lysostaphin (#L7386, Merck, Darmstadt, Germany) and incubated for 15 min at 37 °C. Samples were resuspended in 500 μL TRIzol™ (#15596018, Thermo Fisher Scientific, Waltham, MA, USA) and frozen at −80 °C until they were processed.

For RNA extraction, 200 μL chloroform was added to each sample, after 5 min of centrifugation at maximum speed, the aqueous phase (around 300 μL) was mixed with 70% ethanol and treated with the RNeasy Plus Mini Kit (250) (#74136, QIAGEN) by using a final 40 μL elution volume. Samples were then treated with the TURBO DNA-free kit (#AM1907, Thermo Fisher Scientific, Waltham, MA, USA) according to the manufacturer's instructions for complete digestion of DNA. RNA concentrations were then quantified with NanoDrop 1000 (Thermo Fisher Scientific, Waltham, MA, USA). For reverse-transcription experiments, we used the LunaScript™ RT SuperMix Kit (#E3010L, New England Biolabs, MA, USA) with 150 ng of RNA. Quantitative PCR was performed by using Luna® Universal qPCR Master Mix (#M3003X, New England Biolabs, MA, USA) with 1 μL of cDNA, according to the manufacturer's recommendations. Transcript levels were analyzed using a 7900HT Fast Real-Time PCR System (Applied Biosystems, CA, USA). An efficiency curve was performed for each pair of primers to check specificity and analyze expression. We used the "The Relative Standard Curve Method" for analyzing the qRT-PCR data[74]. The expression of candidate genes was normalized to the housekeeping gene *gyrB*. All measurements were performed at least in triplicates. Primers are listed in Table S2.

### Sialic acid consumption
Bovine submaxillary mucins (BSM) were chemically desialylated for 1 h at 80 °C in a 0.05 M trifluoroacetic acid (TFA) solution. Mucins were then freeze-dried before use.

Bacteria were cultivated in 0.2% chemically desialylated BSM for 4 h and 8 h. After centrifugation at 1700 g for 3 min, free sialic acids in the supernatant were derivatized with DMB (1,2-diamino-4,5-methylene dioxybenzene). Reactions consisted of 7 mM DMB, 18 mM sodium hydrosulfite, 1.4 M acetic acid, and 0.7 M 2-mercaptoethanol and were carried out for 3 h at 50 °C in the dark. DMB−sialic acid derivatives were then quantified by HPLC using a reverse-phase C18 column. A mixture of 84% Milli-Q water, 7% methanol, and 9% acetonitrile was used as an elution buffer at a flow rate of 0.7 mL/min. Detection of

fluorescently labeled sialic acids was achieved at excitation and emission wavelengths of 373 nm and 448 nm, respectively.

For sialic acid consumption of strains ΔnanE, CP-nanE, and CF16 clinical isolates, bacteria were cultivated in CLM supplemented with 1.3 mM Neu5Ac for 4 h and 8 h. After centrifugation at 1700 g for 3 min, free sialic acids in the supernatant were measured directly by using a sialic acid (NANA) colorimetric assay kit (#abx298966-96tests, Abbexa).

## Free sialic acid quantification in sputa
The concentration of free sialic acids in sputa from CF patients was measured using a sialic acid (NANA) colorimetric/fluorometric assay kit (#K566-100, Biovison).

## Evaluation of sialidase activity of isolates
The tested strains were inoculated on chocolate agar polyvitex plates and incubated in a humidified 5% $CO_2$ incubator at 37 °C. *P. melaninogenica* was placed in an anaerobic plastic pouch using Thermo Scientific™ Oxoid™ AnaeroGen™ Compact. After 24 h, the bacteria were scraped and resuspended in PBS to obtain $OD_{600nm} = 1$. Then, 500 µl of bacterial suspension was centrifuged, and the bacteria were resuspended in 100 µl of PBS and 5 µl metapolyzyme (Sigma Aldrich, reconstituted in 750 µL of PBS). Suspensions were incubated at 37 °C for 1 h under constant shaking. The resulting lysates were centrifuged for 5 min (16,000 × g; room temperature (RT)), and supernatants were stored at −80 °C before being subjected to the sialidase activity assay.

Mucins were purified from sputa of CF patients, as previously described[30], before being dissolved to a final concentration of 1 mg/ml in PBS. Mucin solution and bacterial lysates were mixed in a 2:1 ratio and incubated for 3 hours at 37 °C with constant shaking.

The quantification of free sialic acid was measured using an HPLC or highly sensitive sialic acid (NANA) assay kit (ab83375, Abcam).

## O-Glycan release, permethylation, and MALDI-TOF MS
Bovine submaxillary mucins were submitted to β-elimination under reductive conditions (0.1 M KOH, 1 M KBH4 for 24 h at 45 °C). Borate salts were removed by several co-evaporations with methanol before purification on a cation exchange resin column (Dowex 50 × 2, 200–400 mesh, H+ form). The oligosaccharides-alditols fractions were then permethylated, in their anhydrous form, in a solution containing 200 µL dimethylsulfoxide, 300 µL iodomethane and 1 g NaOH, for 2 h, before adding 1 mL acetic acid (5% (v/v)) to stop the reaction. After derivatization, the reaction products were dissolved in 200 µL of methanol and further purified on a C18 Sep-Pak column (Oasis HLB, Waters, Milford, MA, USA).

Permethylated oligosaccharides were analyzed by matrix-assisted laser desorption ionization–time of flight (MALDI-TOF) mass spectrometry (MS) in positive ion reflective mode as [M+Na]+. Samples were dissolved in a methanol/water solvent (50:50) and coated on a MALDI target in addition to a 2,5-dihydroxybenzoic acid (DHB) matrix at a volume/volume dilution.

## Air−liquid interface model and infection
Air−Liquid Interface (ALI) model was used to mimic the physiological environment of lungs through the production of thick mucus[42]. Briefly, Corning® Costar® Transwell® polyester membrane cell culture inserts with a surface area of 0.33 cm$^2$ (corresponding to 24-wells disposition) and pore size of 0.4 µm (#CLS3470, Merck, Darmstadt, Germany) were used to prepare the ALI model. Prior to cell seeding, filter membranes were coated with type IV collagen from the human placenta (#C7521, Merck, Darmstadt, Germany) for 24 h. Cells ($3.0 × 10^5$) were seeded onto the apical chamber in 200 µL of opti-MEM medium, and 1 mL of the same medium was added in the basal chamber (medium is changed on both sides on day 3). On day 5, the apical medium was removed, and only the medium in the basal chamber was changed three times per week. Cells were allowed to grow for 14 to 17 days when mucus was evenly distributed at the surfaces of cell layers.

The release of free sialic acid from airway mucus in the ALI model was evaluated. 5 µL sialidase 0.5 U/mL DPBS/0.1% BSA (Sialidase from *Vibrio cholerae* #11080725001, Sigma-Aldrich Chimie, Germany), 5 µL DPBS/0.1% BSA or 1000 *S. mitis* was added to each filter. Twenty-four hours later, mucus was washed with ringer solution (#1155250001, Merck, Darmstadt, Germany), which allowed mucus to solubilize. 50 µL ringer solution was added on the apical side, filters were incubated at 37 °C for 15 min, and the above operation was repeated twice. The solution was collected and centrifuged at maximum speed for 5 min, the supernatant was collected and free sialic acid concentration was measured by using a sialic acid (NANA) colorimetric/fluorometric assay kit (#K566-100, Biovison).

One day before infection, the opti-MEM medium was replaced by Hank's medium. For infection with the addition of free sialic acid, 2 h prior to infection, 5 µL Neu5Ac (0.4 g/L pH 7.2) or 5 µL $H_2O$ was added to each filter. For infection with the addition of sialidase, 24 h prior to infection, 5 µL sialidase 0.5 U/mL DPBS/0.1% BSA or 5 µL DPBS/0.1% BSA was added to each filter. Strains were grown overnight in BHI. Bacteria were then diluted at 1:1000 into fresh BHI and cultivated until $OD_{600nm}$ ~ 0.6. Then filters were inoculated with 5 µL DPBS suspension containing 100 bacteria and incubated for 16 h at 37 °C.

To evaluate bacterial growth, mucus was washed twice with 50 µL ringer solution as described above. After the centrifugation, the supernatant was removed and the bacterial pellet was suspended in 200 µL DPBS. Finally, 20 µL bacterial solution from each sample was serially diluted in DPBS and spread on BHI Petri dishes.

## ALI model and infection with F508del CFBE41o- cell line
In total, $3.5 × 10^5$ CFBE cells were seeded into the apical chamber. This cell line required 30 days to produce a sufficiently dense mucus layer. Only 10 bacteria are used for inoculation. The remainder of the procedure is conformed to the Calu-3 cell procedure.

## Murine pneumonia infection
Housing conditions: mice were bred and confined in individually ventilated cages (IVCs) with a 12-h light−dark cycle and a temperature range of 20–23 °C and 40–60% humidity in the animal platform.

Overnight cultures of *S. aureus* strains USA300 and ΔagrC grown in BHI medium were diluted to a final $OD_{600nm}$ of 0.1 in 10 ml fresh BHI medium and grown at 37 °C to reach an $OD_{600nm}$ of 0.8. After centrifugation, the bacterial pellet was washed twice with PBS and adjusted to the desired infection inoculum of $2 × 10^8$ CFU/mice. Seven-week-old C57BL/6 J female mice (Janvier-Labs) were anesthetized by inhalation of isoflurane and inoculated intranasally with $2 × 10^8$ CFU in 10 µL PBS or in PBS containing sialic acid. Five mice for each group. Twenty-four hours after infection, mice were sacrificed, and lung tissues were blended and homogenized. Homogenates were serially diluted and plated on BHI agar plates to determine CFU counts.

## RNA sequencing
Total RNA from *S. aureus* WT strain was extracted from 3-mL cultures grown in CLM + Glc or CLM + Neu5Ac until $OD_{600nm}$ ~ 0.6 following the previously described method. RNA quality was assessed using an Agilent Bioanalyzer. RNA sequencing was performed by the platform GENOM'IC of Cochin Institute using TruSeq Stranded Total RNA libraries followed by sequencing on a MiSeq instrument (1 × 150 bp). Bioinformatic pipeline analysis is based on STAR (v2.7.6a) and featureCounts, followed by analysis using R package DESeq2 (v1.26.0).

## *S. aureus* ΔnanE, ΔnanR deletion mutant construction and complementation
The deletion/replacement ΔnanE/npt2, ΔnanR/npt2 mutants of *S. aureus* USA300 strain were obtained by using pMAD. The kanamycin

resistance gene *npt2* with pGro promoter was cloned in pMAD between two DNA fragments corresponding to the chromosomal regions upstream and downstream of the target gene sequence (primers are in Table S3). The constructed plasmids pMAD-*nanE* and pMAD-*nanR* were created using NEBuilder HiFi DNA assembly kit (#E5520S, NEB). The product was transferred into *E. coli* with a standard heat shock protocol. The resulting plasmid was electroporated into RN4220 recipient strain and then transferred to USA300-Ery$^S$. Growth at non-permissive temperature (42 °C) was followed by several subcultures at 30 °C to favor double crossing-over events. The deletion/replacement mutant was verified by PCR using seq-primers (Table S3). To complement these two mutant strains, the plasmid pCN57-CP-*nanE* or pCN57-CP-*nanR* was constructed by homologous recombination. The pCN57 linear vector and target gene fragments were amplified by PCR. To allow recombination, an overlapping region of 20 bp was added at each end. *E. coli* DH5α was chemically transfected with a mix of both PCR products (insert: vector stoichiometry 2:1, 100 ng of the vector) in 10 μL MilliQ water. The colonies containing plasmid were checked by PCR. The complemented plasmid was then transfected into *S. aureus* RN4220 prior to electroporation into *S. aureus* USA300 Δ*nanE* or USA300 Δ*nanR*. The complemented strain CP-*nanE* or CP-*nanR* was finally checked for complementation of wild-type genes by PCR sequencing (Eurofins Genomics) using specific primer pairs listed in Table S3.

### Microbiota composition analysis through 16S rRNA gene sequencing and shotgun metagenomic sequencing

For DNA extraction from sputa samples, 100 μL of each patient's sputum was aliquoted into 1.5-mL Eppendorf tubes, and 10 μL of Metapolyzyme (Sigma Aldrich, reconstituted in 750 μL of PBS) was added. The samples were mixed gently six times, and incubated for 1 h at 37 °C. Then, a DNA extraction kit (Qiagen, DNeasy Blood & Tissue Kit) was used following the manufacturer's protocol. The 16S rRNA genes, region V4, were PCR amplified from each sample using a composite forward primer and a reverse primer containing a unique 12-base barcode, designed using the Golay error-correcting scheme, which was used to tag PCR products from respective samples[75]. We used the forward primer 515 F 5′- *AATGATACGGCGACCACCGAGATCT ACACGCT*XXXXXXXXXXXX**TATGGTAATT*GT***GTGYCAGCMGCCGCGG TAA*−3′: the italicized sequence is the 5′ Illumina adaptor, the 12 X sequence is the golay barcode, the bold sequence is the primer pad, the italicized and bold sequence is the primer linker, and the underlined sequence is the conserved bacterial primer 515 F. The reverse primer 806 R used was 5′-*CAAGCAGAAGACGGCATACGAGAT***AGT CAGCCAGCC***GGACTACNVGGGTWTCTAAT*−3′: the italicized sequence is the 3′ reverse complement sequence of Illumina adaptor, the bold sequence is the primer pad, the italicized and bold sequence is the primer linker and the underlined sequence is the conserved bacterial primer 806 R. PCR reactions consisted of Hot Master PCR mix (Quantabio, Beverly, MA, USA), 0.2 mM of each primer, 10–100 ng template, and reaction conditions were 3 min at 95 °C, followed by 30 cycles of 45 s at 95 °C, 60 s at 50 °C and 90 s at 72 °C on a Biorad thermocycler. PCRs products were purified with Ampure magnetic purification beads (Agencourt, Brea, CA, USA), and visualized by gel electrophoresis. Products were then quantified (BIOTEK Fluorescence Spectrophotometer) using Quant-iT PicoGreen dsDNA assay. A master DNA pool was generated from the purified products in equimolar ratios. The pooled products were quantified using Quant-iT PicoGreen dsDNA assay and then sequenced using an Illumina MiSeq sequencer (2 × 250 bp) at Genom'IC sequencing facility. Obtained 16 S rRNA sequences were analyzed using QIIME2−version 2019[76]. Sequences were demultiplexed and quality filtered using the Dada2 method[77] with QIIME2 default parameters in order to detect and correct Illumina amplicon sequence data, and a table of Qiime 2 artifacts was generated. For taxonomy analysis, features were assigned to operational

taxonomic units (OTUs) with a 99% threshold of pairwise identity to the Greengenes reference database 13_8[78,79].

For shotgun metagenomic sequencing, the Collibri™ PS DNA Library Prep Kits for Illumina Systems (Invitrogen™, A38614196) were used to prepare libraries, which were sequenced on a NextSeq 2000 Illumina instrument (2 × 150 bp) at Genom'IC sequencing facility. Sequence analyses were performed through the Bacterial and Viral Bioinformatics Resource Center (BV-BRC 3.31.12)[80]. Taxonomic classification was carried out by using Kraken2 and Kraken2 Standard Database after the removal of human DNA with Hisat2. The Kraken2 reports were visualized as Sankey diagrams generated with Pavian[81]. For the detection of sialidase genes, an assembly step (SPAdes v3.13.0 option –meta min_contig_coverage_threshold 1 and min_contig_length_threshold 1000)[82] followed by an annotation step (RASTtk[83]) was performed.

### Siderophore activity assay

Siderophore activity in culture supernatants was assayed using chrome azurol S (CAS) by procedures previously described[84]. 150 μl of bacteria culture supernatants at the beginning of the stationary phase were mixed with equal volumes of CAS shuttle solution and incubated at room temperature for 1 h. Corresponding sterile culture media were used as the blank, and the absorbance at 630 nm was determined. Siderophore units were calculated as follows: (A630$_{nm}$ of blank − A630$_{nm}$ of the sample)/A630$_{nm}$ of blank × 100%.

### Statistical analyses

Statistical analyses were performed with GraphPad Prism 8.02, as previously described[85]. The normality of the distribution of whole data sets was assessed using a QQ plot. The homogeneity of variance was tested with the Brown-Forsythe F test. In our data, the distribution of the data for each data set respected the normality law and the variance of the groups was homogeneous. Therefore, multiple comparison analyses were assessed with a one-way ANOVA with Bonferroni's multiple comparisons test, and data were expressed as mean ± SEM (relevant *p* values were reported in the figures). The $H_0$ hypothesis was rejected for a significance level of $p \leq 0.05$.

### Reporting summary

Further information on research design is available in the Nature Portfolio Reporting Summary linked to this article.

## Data availability

The source data underlying all the main Figures and Supplementary Figures are provided as a Source Data file. Sequence data are available at https://doi.org/10.5281/zenodo.8380806. The raw data of the transcriptome can be found in the BioProject PRJNA1036601. The datasets generated and analyzed during this study are provided in Supplementary Data 1 and Data 2. Source data are provided in this paper.

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

## Acknowledgements

Xiongqi Ding was supported by a fellowship from the China Scholarship Council (CSC) n°201904910463. Xiali Fu was supported by a fellowship from the China Scholarship Council (CSC) n°202106300010. Anne Jamet was supported by a grant from ANR-21-CE17-0012-01, and Anne Jamet and Mathieu Coureuil were supported by the IdEx Université de Paris ANR-18-IDEX-0001. Benoit Chassaing's laboratory is supported by a Starting Grant from the European Research Council (ERC) under the European Union's Horizon 2020 research and innovation program (grant agreement No. ERC-2018-StG-804135), an award from the Fondation de l'avenir (AP-RM-21-032), ANR grants EMULBIONT (ANR-21-CE15-0042-01) and DREAM (ANR-20-PAMR-0002) and the national program "Microbiote" from INSERM. The authors thank the Genom'IC platforms (INSERM U1016, Paris, France) for their help. We thank BEI Resources for *S. aureus* USA300-LAC and USA300-LAC Δ*agrC* strains provided. Parts of Figs. 1a and 3d were created using templates from Servier Medical Art (http://smart.servier.com/), licensed under a Creative Common Attribution 3.0 Generic License.

## Author contributions

Conceptualization: A.J., A.C. and M.C.; Methodology: A.J., A.C., M.C., C.R.-M., X.D. and A.L.; Investigation: A.J., A.C., M.C., C.R.-M., X.D., A.L., J.P., M.L.M., I.S.-G., A.F., X.Q., X.F., M.D., D.E., I.D., R.L., B.M., H.R., E.R., C.S., B.C. and C.J.G.D.; Visualization: A.J., A.C., M.C., C.R.-M., X.D. and B.C.; Supervision: A.J., A.C. and M.C.; Writing—original draft: A.J. and X.D.; Writing—review & editing: A.J., A.C., B.C., C.R.-M. and M.C.

## Competing interests

The authors declare no competing interests.
