## [Peer Review File · Nature Communications]

REVIEWER COMMENTS

Reviewer #1 (Remarks to the Author):

Ding et al. "Airway environment drives the selection of quorum sensing mutants and promote Staphylococcus aureus chronic lifestyle"

The lung environment is rich in mucin that carries a high proportion of O-glycosidic sialic acids or N-acetylneuraminic acids (Neu5Ac). Although *S. aureus* doesn't produce a sialidase there are commensal bacteria which have this enzyme thus causing release of sialic acid.

In this study the impact of free sialic acid on *S. aureus* USA300 was investigated.

It was found:

- that agr mutants can better consume sialic acid than the parent strain thus providing a growth advantage;
- a Δ nanE mutant was unable to catabolize sialic acid and was accordingly impaired in growth on sialic acid;
- in agr-mutants the nan-gene expression (sialic acid degradation) was upregulated;
- there was co-occurrence of sialidase-producing bacteria and *S. aureus* seen in the airway of 5 children;
- in a murine pulmonary infections model it was shown that the bacterial load of a agr-mutant was higher than that of the parent strain;
- sialic acid upregulated nan and sbn genes as well as the sRNA isrR, but some of the toxin genes were downregulated;

Major concern:

The structure and presentation of the results is very confusing. The results section is intermixed with many published results, so it is difficult to see what was done here and what was done earlier or by other working groups. The presentation of the results is also irritating in some cases. For example, in Fig. 1G and H they are talking about hemolytic and non-hemolytic variants. In Fig. 1E they are using USA300 and Δ agrC. It looks like the authors are not aware of the phenotype of an agr mutant. This ignorance is evident throughout the paper. In an agr mutant, most secreted toxins are drastically downregulated, such as hla, psms and many others; in contrast, many cell-associated proteins, many of which are adhesins, are upregulated, including protein A, fibrinectin-BP and many others.

This upregulation of adhesins in an agr mutant is also the reason why agr mutants often prevail in tissue infections and chronic infections because they have a colonization advantage.

The fact that in an agr mutant not only the adhesins but also the nan genes are upregulated additionally favors the agr mutant in the colonization of the lung tissue.

Although the authors take samples from CF patients, there is almost no discussion of the interaction of *S. aureus* and *Pseudomonas*, and how *S. aureus* prevails against *P. aeruginosa* (e.g. Biswas, Molecular mechanism of *Staphylococcus* and *Pseudomonas* interaction in CF).

Reviewer #2 (Remarks to the Author):

In this manuscript, the authors investigate the role of sialic acid in promoting *S. aureus* growth in the respiratory tract. The authors examine expression of genes required for sialic acid uptake in agr mutants that arise during CF chronic infections and find increased nan gene expression that corresponds to increased sialic acid uptake in culture, compared to wild type strains. The authors go on to measure sialic acid availability in CF sputum and examine relative abundance of bacteria in sputum samples. The authors also test how addition of exogenous sialidases or sialic acid affects *S. aureus* growth with airway cells and in a mouse model of acute lung infection. Lastly, the authors perform RNAseq of wild-type *S. aureus* in media with sialic acid or glucose, and they investigate a potential regulatory role for NanR in controlling expression of the sRNA *isrR*. The idea that other members of the CF microbiota could liberate sialic acid from mucins, making them available for *S. aureus*, is an intriguing hypothesis. However, there are several issues that need to be addressed:

1. Figure 1GH, The description of this assay in the text (page 5, lines 28-33) did not initially make it clear that this was a competition assay with WT and agr mutant *S. aureus*. This was better clarified in the figure legend. Unless there was selective plating for agrC mutants, it can't be assumed that all hemolytic colonies are due to the original agrC mutant. A control showing the percentage of hemolytic-defective colonies that naturally arise in the wild-type strain grown over 5 days in glucose or neu5AC media would help to address this. The decrease in non-hemolytic colonies from Day 0 to Day 1, prior to non-hemolytic bacteria becoming more abundant in the population, is curious. If the agrC mutant had an advantage over wild-type in the presence of Neu5AC, we may expect to see an immediate growth advantage. These data suggest the agr mutant may be adapting to the presence of Neu5AC in the medium. If there are secondary or compensatory mutations arising in the population should be investigated.

2. Figure 2 data does not identify bacteria present to the species level and at most could suggest a correlation between increased abundance of specific families and free sialic acid, but does not prove causation. Metagenomic analysis to show presence of sialidase genes in samples with increased free sialic acid would be more convincing.

3. Figure 3D, If *S. aureus* can rely on endogenous microbiota in the airways to liberate sialic acid from the host epithelium, it should not require addition of exogenous sialic acid to see an increase in bacterial burden. A co-infection with a sialidase producing organism versus a co-infection with a bacterial strain that cannot liberate sialic acid, or a competition assay with a way to identify WT and agr mutant populations, would be convincing here.

4. Figure 4, RNA seq experiments with wild-type *S. aureus* showed an increase in nan gene expression on Neu5AC, but an advantage for wild-type *S. aureus* with sialic acid was not observed in other experiments shown here. Investigating differences in gene expression in an agr mutant background or in the adapted clinical isolates would be informative and help to support the authors earlier claims. The finding that the isrR sRNA and sbn expression are regulated by NanR is interesting, but additional work needs to be done to investigate if iron acquisition or uptake is truly affected in these strains before this claim can be made, including measuring protein levels of the Sbn, or growth or survival of mutants in the presence or absence of iron.

Reviewer #3 (Remarks to the Author):

This is a well written paper with interesting findings. However, to connect this mechanism to CF lung disease, some more work needs to be done.

Major Comments:

1. The data collected from the CF sputum samples is very convincing. However, it would be helpful to know if the increase in free sialic acid precedes Staph infection. Therefore, additional samples from patients not identified as being *S. aureus* positive would be beneficial.
2. The ALI experiments are very interesting, but should be conducted in a cell line with the CFTR mutation, preferably primary cells. The increased mucus present in the airway, postulated as one of the environmental triggers for this interaction, is less likely to be present on normal Calu3 cells.
3. Similarly, the mouse phenotype is very interesting, although adding free sialic acid may not accurately represent the availability of this product in the lung environment. More sophisticated in vivo experiments, such as using the CFTR^{-/-} or Scnn1Tg mouse, which would have more mucins present in the lung, or preseeded the lung with the proposed sialidase-producing bacteria, would be more convincing.

Minor:

1. Figure 2 is very difficult to read. From the way A and B are presented, it looks like there are changes between native mucins and mucins with USA300. his figure needs to be enlarged at minimum.

REVIEWER COMMENTS

Reviewer #1 (Remarks to the Author):

Ding et al. "Airway environment drives the selection of quorum sensing mutants and promote *Staphylococcus aureus* chronic lifestyle"

The lung environment is rich in mucin that carries a high proportion of O-glycosidic sialic acids or N-acetylneuraminic acids (Neu5Ac). Although *S. aureus* doesn't produce a sialidase there are commensal bacteria which have this enzyme thus causing release of sialic acid. In this study the impact of free sialic acid on *S. aureus* USA300 was investigated. It was found:

- that *agr* mutants can better consume sialic acid than the parent strain thus providing a growth advantage;
- a Δ nanE mutant was unable to catabolize sialic acid and was accordingly impaired in growth on sialic acid;
- in *agr*-mutants the *nan*-gene expression (sialic acid degradation) was upregulated;
- there was co-occurrence of sialidase-producing bacteria and *S. aureus* seen in the airway of 5 children;
- in a murine pulmonary infections model it was shown that the bacterial load of a *agr*-mutant was higher than that of the parent strain;
- sialic acid upregulated *nan* and *sbm* genes as well as the sRNA *isrR*, but some of the toxin genes were downregulated;

Major concern:

The structure and presentation of the results is very confusing. The results section is intermixed with many published results, so it is difficult to see what was done here and what was done earlier or by other working groups. The presentation of the results is also irritating in some cases. For example, in Fig. 1G and H they are talking about hemolytic and non-hemolytic variants. In Fig. 1E they are using USA300 and Δ agrC. It looks like the authors are not aware of the phenotype of an *agr* mutant.

1- We understand that the presentation and legending of the results might have been confusing. In particular, the choice of referring to the "non-hemolytic phenotype" of the *agr* mutants in Fig 1G and 1H needed to be clarified:

- When we analyzed gene expression (such as in Fig 1C) or ability to consume sialic acid (such as in Fig 1E), we used the name of the tested strain (USA300-LAC wild-type and its Δ agrC derivative).
- In contrast, in the competition assay (such as in Fig 1G and H), we chose to use the hemolytic phenotype of the colonies allowing an easy quantification of the proportion of *agr* mutants and wild-type colonies on the plates (*agr* mutants were non-hemolytic whereas wild-type were hemolytic). Differences in colony aspects are shown in supplementary Fig S2C.
- In addition, selective plating on TS plates with or without antibiotic (erythromycin) has been performed and showed a similar proportion of antibiotic resistant/susceptible colonies compared to non-hemolytic/hemolytic colonies (supplementary Fig S2D). The legend of Fig 1 has been updated.

[...] In an *agr* mutant, most secreted toxins are drastically downregulated, such as *hla*, *psms* and many others; in contrast, many cell-associated proteins, many of which are adhesins, are upregulated, including protein A, fibrinectin-BP and many others. This upregulation of adhesins in an *agr* mutant is also the reason why *agr* mutants often prevail in tissue infections and chronic infections because they have a colonization advantage. The fact that in an *agr* mutant not only the adhesins but also the *nan* genes are upregulated additionally favors the *agr* mutant in the colonization of the lung tissue.

2- Of course other well-known factors such as upregulation of adhesins in *agr* mutants contribute to explain their ability to establish chronic infections. We recalled it in the text (see lines 297-299).

One major finding of our study is to demonstrate that free sialic acid, which is widely available from airway mucins, represents an overlooked metabolic factor with an unanticipated role in driving the selection of *agr* variants. We now insist on this point in the introduction, lines 91-94.

Although the authors take samples from CF patients, there is almost no discussion of the interaction of *S. aureus* and *Pseudomonas*, and how *S. aureus* prevails against *P. aeruginosa* (e.g. Biswas, Molecular mechanism of *Staphylococcus* and *Pseudomonas* interaction in CF).

3- As suggested, the interaction between *S. aureus* and *P. aeruginosa* in the context of CF patient was briefly discussed in the amended manuscript and the suggested citation added lines 310-318.

Reviewer #2 (Remarks to the Author):

In this manuscript, the authors investigate the role of sialic acid in promoting *S. aureus* growth in the respiratory tract. The authors examine expression of genes required for sialic acid uptake in *agr* mutants that arise during CF chronic infections and find increased *nan* gene expression that corresponds to increased sialic acid uptake in culture, compared to wild type strains. The authors go on to measure sialic acid availability in CF sputum and examine relative abundance of bacteria in sputum samples. The authors also test how addition of exogenous sialidases or sialic acid affects *S. aureus* growth with airway cells and in a mouse model of acute lung infection. Lastly, the authors perform RNAseq of wild-type *S. aureus* in media with sialic acid or glucose, and they investigate a potential regulatory role for NanR in controlling expression of the sRNA *isrR*. The idea that other members of the CF microbiota could liberate sialic acid from mucins, making them available for *S. aureus*, is an intriguing hypothesis. However, there are several issues that need to be addressed:

1. Figure 1GH, the description of this assay in the text (page 5, lines 28-33) did not initially make it clear that this was a competition assay with WT and *agr* mutant *S. aureus*. This was better clarified in the figure legend. Unless there was selective plating for *agrC* mutants, it can't be assumed that all hemolytic colonies are due to the original *agrC* mutant. A control showing the percentage of hemolytic-defective colonies that naturally arise in the wild-type strain grown over 5 days in glucose or Neu5AC media would help to address this. The decrease in non-hemolytic colonies from Day 0 to Day 1, prior to non-hemolytic bacteria becoming more abundant in the population, is curious. If the *agrC* mutant had an advantage over wild-type in the presence of Neu5AC, we may expect to see an immediate growth advantage. These data suggest the *agr* mutant may be adapting to the presence of Neu5AC in the medium. If there are secondary or compensatory mutations arising in the population should be investigated.

- We improved the description of the assay (lines 135-139) for clarification (see also answer to R#1). Selective plating was performed showing the same ratios of non-hemolytic/hemolytic colonies and erythromycin-resistant/susceptible colonies suggesting that non-hemolytic colonies almost exclusively correspond to the *agrC* mutant initially inoculated. This information has been added (lines 142-145).

- As proposed, we performed a control experiment with the wild-type strain grown over 5 days in glucose or Neu5AC media. We did not observe the appearance of hemolytic-defective colonies in any of both conditions suggesting that sialic acid does not impose a pressure leading to the emergence of novel *agr* variants. This information has been added (lines 145-149).

- We checked by whole-genome sequencing the appearance of mutations during the competition assay by sequencing 10 non-hemolytic and 10 hemolytic isolates at day 5 taken from 2 independent experiments. As expected, we detected mutations by comparing day 5-isolates with parental genomes. We did not find any convergent mutations present only in the *agrC* strain compared to the wild-type strain that would explain the selection of *agrC* variants.

Hence, these data suggest the *agr* mutant may be adapting to the presence of Neu5AC in the medium at day 1.

These data are provided in the revised manuscript (lines 149-156 and Dataset S1).

2. Figure 2 data does not identify bacteria present to the species level and at most could suggest a correlation between increased abundance of specific families and free sialic acid, but does not prove causation. Metagenomic analysis to show presence of sialidase genes in samples with increased free sialic acid would be more convincing.

We agreed with R#2 that precise identification of sialidase-producing species was required to strengthen our findings, therefore we conducted the following experiments.

- In order to identify bacteria at the species level, we performed shotgun metagenomic analysis of the 5 sputa (Fig S4B). This analysis showed the presence of species for which sialidase activity has been previously demonstrated (10.1128/jcm.27.1.182-184.1989, 0095-1137/91/091955-04\$02.00/0, 0095-1137/90/061431-03\$02.00/0). Indeed, all the patients harbored at least one of the following species: *Schaalia odontolytica* (previously known as *Actinomyces odontolyticus*), *Prevotella melaninogenica*, or *Streptococcus mitis* group species (*S. mitis*, *S. oralis*, or *S. infantis*).

- Beyond species identification, we detected fragments of sialidase genes in 3 samples. It should be pointed out that due to the high number of human cells within CF sputa (accounting for more than 90% of the raw sequences generated), shotgun metagenomics is not a sensitive technique for the identification of specific bacterial genes.

- In addition, to directly prove that isolates from CF sputa display functional sialidase activity, we isolated 3 bacterial isolates from CF sputa belonging to 3 distinct species (*Prevotella melaninogenica*, *Schaalia odontolyticus* and *Streptococcus infantis*) and we directly confirmed their sialidase activity (Fig 2C).

Collectively, these data directly show that sialidase-producing species are present in CF sputa.

All these new results are now shown in figures 2 and S4 and described lines 170-191.

3. Figure 3D, If *S. aureus* can rely on endogenous microbiota in the airways to liberate sialic acid from the host epithelium, it should not require addition of exogenous sialic acid to see an increase in bacterial burden. A co-infection with a sialidase producing organism versus a co-infection with a bacterial strain that cannot liberate sialic acid, or a competition assay with a way to identify WT and agr mutant populations, would be convincing here.

Indeed, the mouse lung model is imperfect, which has been also highlighted by R#3. In the mouse lung, the mucus layer is very thin and does not recapitulate well the lung of CF patients (10.1152/ajplung.00485.2019, 10.1165/rcmb.2004-0060OC). In line with literature, we confirmed that, in our hands, mucins or free sialic acid were below the detection limit in bronchoalveolar lavages of C57BL/6J mice. We also performed mouse lung infection with a *Streptococcus mitis* strain for which we previously showed its ability to release free sialic (Fig 3A). However, this strain was not able to establish an infection in the lungs of C57BL/6J mice and accordingly, free sialic acid was not detectable in the mouse lungs 24h after infection. For all these reasons, we decided to directly add sialic acid as a proof of concept that sialic acid is sufficient to promote an increase in bacterial burden.

Limitations of the mouse model are now discussed lines 352-356.

4. Figure 4, RNA seq experiments with wild-type *S. aureus* showed an increase in nan gene expression on Neu5AC, but an advantage for wild-type *S. aureus* with sialic acid was not observed in other experiments shown here. Investigating differences in gene expression in an agr mutant background or in the adapted clinical isolates would be informative and help to support the authors earlier claims. The finding that the isrR sRNA and sbn expression are regulated by NanR is interesting, but additional work needs to be done to investigate if iron acquisition or uptake is truly affected in these strains before this claim can be made, including measuring protein levels of the Sbn, or growth or survival of mutants in the presence or absence of iron.

- Indeed, the induction of the expression of the whole *sbn* locus encoding the staphyloferrin B siderophore in wild-type *S. aureus* in presence of sialic acid is one of the main findings of the RNAseq experiment. We have now checked whether sialic acid also has an impact on *sbn* expression in the agr mutant background and have shown that sialic acid similarly induces an increase in *sbn* expression in the agr mutant.

- To verify if the changes in *sbn* expression in WT, agr and nanR mutants correlated with increased siderophore activity, we used chrome azurol S (CAS) phenotypic assay. Our results showed that siderophore activity is indeed increased in WT and agr backgrounds in presence of sialic acid compared to glucose, and in nanR mutant compared to the parental strain.

Altogether, these results demonstrated that sialic acid presence triggers an increased siderophore activity in WT and agr mutant, which is linked to NanR derepression.

These new results have been added (Fig 4E and 4I) and discussed lines 256-259, 283-285.

Reviewer #3 (Remarks to the Author):

This is a well written paper with interesting findings. However, to connect this mechanism to CF lung disease, some more work needs to be done.

Major Comments:

1. The data collected from the CF sputum samples is very convincing. However, it would be helpful to know if the increase in free sialic acid precedes Staph infection. Therefore, additional samples from patients not identified as being *S. aureus* positive would be beneficial.

- Our hypothesis, based on the published studies on airway microbiota (10.1016/j.tim.2023.04.006), was that sialidase-producing species were present regardless of the *S. aureus* presence.

- We managed to obtain 4 sputa of pediatric patients (P21-37, P27-33, P24-20, and P21-25) who were not infected with *S. aureus* and demonstrated the presence of free sialic acid in the 4 CF sputa (Fig S4A).

- In addition, we carried out 16S microbiota (Fig S4C) and found that, similarly to what was observed in patients with *S. aureus*, Streptococcae constituted a major family of the lung microbiota of those CF patients without *S. aureus*. Therefore, we performed MALDI-TOF identification of bacterial isolates from the 4 CF sputa on Columbia sheep agar plates and we were able to identify Streptococcus mitis group species (*S. oralis*). We then confirmed the ability of these 4 isolates to exhibit sialidase activity (Fig S4D).

Collectively (see also response to Q2 of Reviewer#2), our results suggest that sialidase producing species can be found in the airway of CF patients regardless of the presence of *S. aureus*.

These new results have been added (Fig S4).

2. The ALI experiments are very interesting, but should be conducted in a cell line with the CFTR mutation, preferably primary cells. The increased mucus present in the airway, postulated as one of the environmental triggers for this interaction, is less likely to be present on normal Calu3 cells.

We now provide the results of the ALI experiments with a human CF bronchial epithelial cell line with the CFTR mutation (homozygous F508del CFBE410-). We confirmed that presence of free sialic acid in the mucus favored isolates harboring an Agr system inactivation.

These new results have been added (Fig S5).

3. Similarly, the mouse phenotype is very interesting, although adding free sialic acid may not accurately represent the availability of this product in the lung environment. More sophisticated in vivo experiments, such as using the CFTR^{-/-} or Scnn1Tg mouse, which would have more mucins present in the lung, or preseeded the lung with the proposed sialidase-producing bacteria, would be more convincing.

Indeed, the mouse lung model is imperfect, which has been also highlighted by R#2. In the mouse lung, the mucus layer is very thin and does not recapitulate well the lung of CF patients (10.1152/ajplung.00485.2019, 10.1165/rcmb.2004-0060OC). In line with literature, we confirmed that, in our hands, mucins or free sialic acid were below the detection limit in bronchoalveolar lavages of C57BL/6J mice. We also performed mouse lung infection with a *Streptococcus mitis* strain for which we previously showed its ability to release free sialic (Fig 3A). However, this strain was not able to establish an infection in the lungs of C57BL/6J mice and accordingly, free sialic acid was not detectable in the mouse lungs 24h after infection. Currently, we do not have the CFTR^{-/-} or Scnn1Tg mice and cannot use them due to the lack of ethical authorization for these specific mice.

For all these reasons, we decided to directly add sialic acid as a proof of concept that sialic acid is sufficient to promote an increase in bacterial burden.

Limitations of the mouse model are now discussed lines 352-356.

Minor:

1. Figure 2 is very difficult to read. From the way A and B are presented, it looks like there are changes between native mucins and mucins with USA300. This figure needs to be enlarged at minimum.

We improved the quality of the presentation by enlarging the MS spectra shown and adding a table detailing change in the proportions of oligosaccharide structures. Overall, these results highlight that the changes observed are not related to sialic acid release, which is consistent with the absence of sialidase activity in *S. aureus*. Indeed, around 86% of O-glycans were sialylated in native mucins and the same level of sialylation (around 88%) were observed after incubation of mucins with USA300-LAC (Fig S3 and lines 163-166).

REVIEWERS' COMMENTS

Reviewer #2 (Remarks to the Author):

The authors were responsive to my critiques on their earlier draft, and the revised manuscript includes new data that addresses my initial concerns.

Experiments showing that agr variants do not arise and clarifying the competition assay with selective plating strengthened Figure 1 and addressed my concerns regarding this work.

The additional metagenomics analysis performed to identify sialidase producing species and sialidase genes addressed my concerns regarding Figure 2 and is a good addition to the study, to confirm this metabolic capability is present in patient samples.

The limitations of the mouse model and availability of sialic acid regarding Fig. 3 are understood, and the authors' response to discuss these limitations is sufficient to address this concern.

The new experiments investigating siderophore production in the presence of sialic acid that have been added to Figure 4 provided useful data and address my earlier concerns with validation of the RNA seq conclusions.

Reviewer #3 (Remarks to the Author):

The additional experiments have improved the manuscript significantly, and this reviewer feels that the comments were all appropriately addressed.

Reviewer #4 (Remarks to the Author):

In this study, the authors characterize sialic acid utilization in *S aureus* USA300 and an agr mutant, as well as in two pairs of CF early and late isolates. They showed the agr mutant and long term adapted CF

isolates (also agr mutants) had increase ability to utilize sialic acid in different in vitro settings and in vitro in a murine lung infection model (supplemented with exogenous sialic acid). These finding reveal a new carbon source for S aureus in the CF airways and this may represent a potential selection pressure for host adaptation.

Some considerations for the authors to address.

lines 121-123 The authors show that this increased sialic utilization is a feature of "two prototypical Agr dysfunctional ... CF late isolates". It would be important to know if this is true of most or all late CF isolates of S aureus to make any general statements about this as an adaptation to the CF lungs. It could be assumed that this is an indirect consequence of selection for agr mutants. If sialic acid utilization was the primary selection, one would expect nanR mutations to be prominent. It is likely that there is strong selection for down regulation of agr for many reasons, sialic acid utilization may be one of them. There needs to be a bit more nuanced discussion around this.

Fig 1C,D - it does appear that the CF strains have much lower expression than the USA300 wild type strain although this is confounded by using "post exponential phase" cells for gene expression. It seems this would have been more relevant at 4 or even 8 hrs.

Fig 1H - the result here is somewhat confusing as well. Given that the agr mutant has even a modest better consumption of sialic acid than the parent strain, shouldn't the ratios be similar or slightly more at Day 1? The increase proportion over several days seems to indicate something else may be going on. This may related to increase survival of the mutant - which may or may not be dependent on sialic acid consumption. Rather than the ratios, the authors should plot absolute CFUs over time. This seems more complicated than just sialic acid utilization.

The carbon limited medium (CLM) is supplemented with 0.25% caseamino acids which would be an additional carbon source (although the nanE mutant does not grow in CLM with sialic acid supplementation). The statement in the discussion lines 293-4, maybe a bit too strong is if this is slower death rather than better growth.

The growth data in the ALI model and in the sialic acid supplement lung infection model are consistent the in vitro data. Given the marked reduction in virulence gene expression in the presence of Sialic acid, it would have been useful to see a there was a corresponding affect on pathogenicity in these models (eg increase how cell death in the on vitro system, altered host response (cytokines) or inflammation in the murine model.

line 286-289 - It would be nice to see to close this part of the story, but not essential, to show direct repression of isrR by nanR using EMSAs which have been established previously for the binding the nan promoters.

REVIEWER COMMENTS

Reviewer #2 (Remarks to the Author):

The authors were responsive to my critiques on their earlier draft, and the revised manuscript includes new data that address my initial concerns.

Experiments showing that agr variants do not arise and clarifying the competition assay with selective plating strengthened Figure 1 and addressed my concerns regarding this work.

The additional metagenomics analysis performed to identify sialidase producing species and sialidase genes addressed my concerns regarding Figure 2 and is a good addition to the study, to confirm this metabolic capability is present in patient samples.

The limitations of the mouse model and availability of sialic acid regarding Fig. 3 are understood, and the authors' response to discuss these limitations is sufficient to address this concern.

The new experiments investigating siderophore production in the presence of sialic acid that have been added to Figure 4 provided useful data and address my earlier concerns with validation of the RNA seq conclusions.

- On behalf of all the authors, we would like to thank the reviewer #2 for all the constructive comments that helped us improve our manuscript.

Reviewer #3 (Remarks to the Author):

The additional experiments have improved the manuscript significantly, and this reviewer feels that the comments were all appropriately addressed.

- We would like to thank the reviewer #3 for the positive comments and valuable suggestions to improve the quality of our manuscript.

Reviewer #4 (Remarks to the Author):

In this study, the authors characterize sialic acid utilization in *S aureus* USA300 and an agr mutant, as well as in two pairs of CF early and late isolates. They showed the agr mutant and long term adapted CF isolates (also agr mutants) had increase ability to utilize sialic acid in different in vitro settings and in vitro in a murine lung infection model (supplemented with exogenous sialic acid). These finding reveal a new carbon source for *S aureus* in the CF airways and this may represent a potential selection pressure for host adaptation.

Some considerations for the authors to address.

lines 121-123 The authors show that this increased sialic utilization is a feature of "two prototypical Agr dysfunctional ... CF late isolates". It would be important to know if this is true of most or all late CF isolates of *S aureus* to make any general statements about this as an adaptation to the CF lungs. It could be assumed that this is an indirect consequence of selection for agr mutants. If sialic acid utilization was the primary selection, one would expect nanR mutations to be prominent. It is likely that there is strong selection for down regulation of agr for many reasons, sialic acid utilization may be one of them. There needs to be a bit more nuanced discussion around this.

We agree with the reviewer that selection of Agr dysfunctional strains is probably driven by the many advantages derived from the deregulation of the agr regulon. Indeed, the consequences of Agr dysfunction include increased biofilm formation and major metabolic changes. The fact that sialic acid is available in the CF lung and that prototypic agr mutants (in USA300-LAC reference strain and 2 clinical isolates) are able to metabolize it more rapidly suggest that sialic acid is a likely contributor to the selection of agr mutants. We agree that it would be interesting to extend our observations to a larger number of lung-adapted isolates, with or without agr dysfunction, in future studies.

Fig 1C,D - it does appear that the CF strains have much lower expression than the USA300 wild type strain although this is confounded by using "post exponential phase" cells for gene expression. It seems this would have been more relevant at 4 or even 8 hrs.

- It has been shown that the Agr regulon and the fold changes in gene expression upon *agr* inactivation are highly variable between different lineages (doi:10.1128/IAI.00046-11). We are therefore not surprised to observe differences in gene expression between CF and USA300 strains.

- During the late stationary phase, the medium became depleted in sialic acid, which was consumed by the bacteria. Since the presence of sialic acid is necessary to induce *nan* gene expression, we focused on the post-exponential phase of growth, where we observed the most significant difference in *nan* gene expression.

Fig 1H - the result here is somewhat confusing as well. Given that the *agr* mutant has even a modest better consumption of sialic acid than the parent strain, shouldn't the ratios be similar or slightly more at Day 1? The increase proportion over several days seems to indicate something else may be going on. This may related to increase survival of the mutant - which may or may not be dependent on sialic acid consumption. Rather than the ratios, the authors should plot absolute CFUs over time. This seems more complicated than just sialic acid utilization.

The carbon limited medium (CLM) is supplemented with 0.25% caseamino acids which would be an additional carbon source (although the *nanE* mutant does not grow in CLM with sialic acid supplementation). The statement in the discussion lines 293-4, maybe a bit too strong is if this is slower death rather than better growth. We have modified the statement to make it less strong.

- In this experiment, we maintained identical dilutions each day, resulting in minimal variation in total bacterial numbers. From day 3 onwards, the *agrC* mutant strain consistently exhibited a higher CFU number than the USA300 strain. On the basis of the data currently available (RNA-seq and genome sequencing), we have no hypotheses other than the increased capacity to consume sialic acid. We agree that the decrease on day 1 is intriguing and should be studied in more detail.

The growth data in the ALI model and in the sialic acid supplement lung infection model are consistent the in vitro data. Given the marked reduction in virulence gene expression in the presence of Sialic acid, it would have been useful to see a there was a corresponding effect on pathogenicity in these models (eg increase how cell death in the on vitro system, altered host response (cytokines) or inflammation in the murine model.

- We agree that understanding the role of sialic acid in the regulation of virulence is a worthy avenue of research. In addition, the therapeutic potential of sialic acid-containing compounds should be explored in future studies.

line 286-289 - It would be nice to see to close this part of the story, but not essential, to show direct repression of *isrR* by *nanR* using EMSAs which have been established previously for the binding the *nan* promoters.

- We agree that it would be interesting to conduct a second study focusing on the NanR regulon and that EMSA would in this context provide definitive evidence for a direct regulation of *isrR* by NanR.